# Dietary Complex and Slow Digestive Carbohydrates Promote Bone Mass and Improve Bone Microarchitecture during Catch-Up Growth in Rats

**DOI:** 10.3390/nu14061303

**Published:** 2022-03-19

**Authors:** Pilar Bueno-Vargas, Manuel Manzano, Íñigo M. Pérez-Castillo, Ricardo Rueda, José M. López-Pedrosa

**Affiliations:** Abbott Nutrition R&D, Abbott Laboratories, 18004 Granada, Spain; manuel.manzano@abbott.com (M.M.); inigomaria.perez@abbott.com (Í.M.P.-C.); ricardo.rueda@abbott.com (R.R.); jose.m.lopez@abbott.com (J.M.L.-P.)

**Keywords:** catch-up growth, bone formation, peak bone mass, slow digestible carbohydrates, bone microstructure

## Abstract

Catch-up growth is a process that promotes weight and height gains to recover normal growth patterns after a transient period of growth inhibition. Accelerated infant growth is associated with reduced bone mass and quality characterized by poor bone mineral density (BMD), content (BMC), and impaired microarchitecture. The present study evaluated the effects of a diet containing slow (SDC) or rapid (RDC) digestible carbohydrates on bone quality parameters during the catch-up growth period in a model of diet-induced stunted rats. The food restriction period negatively impacted BMD, BMC, and microarchitecture of appendicular and axial bones. The SDC diet was shown to improve BMD and BMC of appendicular and axial bones after a four-week refeeding period in comparison with the RDC diet. In the same line, the micro-CT analysis revealed that the trabecular microarchitecture of tibiae and vertebrae was positively impacted by the dietary intervention with SDC compared to RDC. Furthermore, features of the cortical microstructure of vertebra bones were also improved in the SDC group animals. Similarly, animals allocated to the SDC diet displayed modest improvements in growth plate thickness, surface, and volume compared to the RDC group. Diets containing the described SDC blend might contribute to an adequate bone formation during catch-up growth thus increasing peak bone mass, which could be linked to reduced fracture risk later in life in individuals undergoing transient undernutrition during early life.

## 1. Introduction

Adequate linear growth is a major determinant of childhood well-being. Poor health conditions and undernutrition can compromise optimal growth patterns [1]. Stunting and linear growth retardation lead to delayed child development and are associated with negative effects on overall bone mass, characterized by reduced bone mineral density (BMD) and content (BMC); and bone quality, denoted by impaired microarchitecture. Impaired bone mass and quality can translate into early onset osteopenia and osteoporosis, and a higher risk of bone fractures [2,3,4]. In this sense, causes of suboptimal lineal growth arise from different factors including prenatal maternal undernutrition, recurrent infections, exposure to toxins, and intestinal disorders [3]. Nonetheless, although stunting does not always indicate undernutrition, it is undeniable that a deficient nutritional status during the early stages of life constitutes a key contributor to growth retardation and to its short- and long-term consequences [5]. 

Catch-up growth is defined as “height velocity above the statistical limits of normal for age and/or maturity during a defined period, following a transient period of growth inhibition” [6]. It constitutes a process that enables weight and height recovery after a period of impaired growth. In this regard, dietary interventions have been proposed for the promotion of catch-up growth in malnourished children. However, a poor dietary intervention can lead to accelerated infant growth, which has been linked to an increased risk of obesity and cardiovascular disease [7]. Furthermore, catch-up growth induced by an inadequate nutrition might be insufficient to promote optimal bone health, which would lead to an increased risk of bone fractures later in life [8,9,10]. Accordingly, different experimental animal models of stunting followed by a period of diet-induced catch-up growth have been explored. These study designs consist of restricting the protein-calorie intake of animals for a fixed period of time to later evaluate the impact that a refeeding period has on growth and bone-related outcomes compared to *ad libitum* fed animals [9]. Interestingly, nutritional interventions with different dietary components have been shown to exert disparate effects on skeletal growth. In this respect, studies conducted in animal models of food restriction have shown that the replacement of the type of proteins (i.e., replacing whey protein by casein) significantly improves aspects of bone quality after the period of refeeding [11]. Similarly, subtle modifications in the fatty acid composition of catch-up diets may improve bone elongation and microstructure [12]. Hence, modulating the composition of dietary interventions during catch-up growth might be beneficial for children with growth disorders. 

Besides lipids and proteins, carbohydrates (CHO) constitute one of the main sources of energy and are essential for growth and development. In this regard, not only the quantity but also the quality of CHO presented in diets is important, since a nutrition consisting of carbohydrates with a high glycemic index, as a measure of their quality, elicits a higher glycemic response that could impair bone metabolism. Hence, rapid digestible carbohydrates (RDC), which deliver a faster glycemic response, might negatively affect bone health [13,14]. On the other hand, slow digestible carbohydrates (SDC) induce a more sustained glycemic response characterized by a less pronounced postprandial rise in blood glucose and insulin [15,16]. Notably, hyperglycemia has been linked to reduced osteoblast numbers and function along with inhibited osteoblast maturation and bone mineralization [13,14]. Thus, SDC might support bone growth and development during the catch-up period. Moreover, low glycemic index diets have been demonstrated to confer positive effects on metabolic disorders linked to an unhealthy catch-up growth such as obesity, diabetes mellitus, and cardiovascular disease [17,18,19]. In a previous work, we provided evidence that the consumption of an SDC diet promotes a healthier catch-up growth phenotype regarding adipose tissue, muscle, and liver, which was characterized by a more efficient utilization of glucose, increased insulin sensitivity, and reduced lipogenic metabolism when compared to RDC [20]. Furthermore, the consumption of the SDC diet led to enhanced muscle differentiation and improved fuel utilization [20]. Based on the results obtained in our previous work, SDC diet positively influenced muscle function, which suggests that they might also exert positive effects on bone development during catch-up growth [21], as bone development is driven by strain from muscle force, and therefore, by muscle development [22,23]. 

In light of our previous study, we aimed to evaluate the effects that a nutritional intervention containing slow digestible carbohydrates has on BMD, BMC, growth plate, and bone microarchitecture of rats undergoing catch-up growth after a period of dietary restriction compared to the administration of rapid digestible carbohydrates. In the present study, we hypothesized that the consumption of an SDC-containing diet could lead to improved bone mass and microarchitecture parameters of rats compared to a RDC diet. 

## 2. Materials and Methods

### 2.1. Housing

Experimental procedures of the present research were conducted in strict accordance with the ethical guidelines for animal experimentation provided by the Spanish National Research Council (RD 53/2013 1 February) (approval code 23/05/2016/088), and in full agreement with the “Directive 2010/63/EU of the European Parliament and of the Council on the protection of animals used for scientific purposes”. 

Forty-two weanling male Sprague Dawley rats (Oncin France strain A (OFA); 21–25 days old) were provided by Charles Rivers (Orleans, France). Animals were individually housed in cages and kept under housed standard environmental conditions (22 °C, relative humidity of 50%, 12 h light/dark cycle). 

### 2.2. Experimental Procedures

Rats were randomly assigned to two groups: a non-restricted rats group (NR, *n* = 16) and a restricted rats group (RR, *n* = 26). NR rats were fed to standard reference diet for growing rodents (AIN93G: 7.0% fat, 17.8% protein, 67.4% carbohydrates, and 4.8% fiber per weight) *ad libitum*, while rats in the RR group received 70% the amount of food received by NR the previous day corrected by body weight (food intake in g/100 g body weight per day). Restriction and refeeding periods lasted four weeks each in one. A more detailed description of the experimental procedures is reported elsewhere [20].

After the four-week restriction period, RR rats were randomly allocated to two different experimental diets (*n* = 10 animals per group), while NR rats received the standardized rodent diet for maintenance (AIN93M). The remaining subset of animals from NR and RR groups (*n* = 6 animals per group) were sacrificed at the end of the restriction period to assess the impact of the nutritional restriction on BMD, BMC, and trabecular microstructure of isolated bones (femurs, tibiae, and lumbar vertebrae (LV 2–5)). 

All the diets were formulated with the requirements of vitamins and minerals based on the standardized rodent diet. All animals were fed *ad libitum* to the diets and were given free access to de-ionized water during the refeeding period. Experimental diets were formulated either with a rapid digestible carbohydrates blend (RDC group) or a slow digestible carbohydrates blend (SDC group), which provided them with a different glycemic load (GL) index. The GL was calculated as the sum of the individual contributions of each CHO contained in the diet, which was obtained by multiplying each carbohydrate content by their glycemic index using glucose as reference food. Hence, RDC presented a GL index of 851, while the GL index of the SDC diet was 468. The composition of control and experimental diets is presented in Table 1.

At the end of the four-week refeeding period, rats were sacrificed by exsanguination under intraperitoneal anesthesia in post-absorptive conditions (one hour after oral meal tolerance challenge with the experimental diets; 10 kcal diet/kg body weight (BW)). Blood samples were collected into serum tubes after oral gavage and serum was isolated after centrifugation at 1500× *g* for 10 min at 4 °C, frozen, and kept at −80 °C for posterior analysis. Femurs, tibiae, and lumbar vertebrae (LV 2–5) were isolated and kept at −20 °C until analysis. 

### 2.3. Biochemical Parameteres

Minerals (calcium, magnesium, and phosphorus) and alkaline phosphatase concentration were quantified in serum by colorimetric assay using a Pentra 400 clinical chemistry analyzer (Horiba ABX, Montpellier, France). Briefly, magnesium was analyzed using the xylidyl blue method, while calcium was determined using the arsenazo III assay. Similarly, phosphorus was measured using the phosphomolybdate reaction assay, and alkaline phosphatase was quantified using the p-nitrophenylphosphate method. Serum hormone levels were assessed using the Milliplex Map Rat Bone Magnetic Bead Panel kit and the Milliplex Map Rat Metabolic Magnetic Bead Panel kit (Millipore; Billerica, MA, USA) measured by Bio-Plex^®^ 200 (Bio-Rad Laboratories, Inc., Hercules, CA, USA). Analyzed serum hormones consisted of osteoprotegerin (OPG), parathyroid hormone (PTH), insulin, leptin, growth hormone (GH), luteinizing hormone (LH), and follicle-stimulating hormone (FSH). Assays were performed in a 96-well plate following product instructions. 

### 2.4. Ex Vivo Densitometry Analysis

Bone mineral density (BMD) and content (BMC) of isolated bones (femurs, tibiae, and vertebrae) as well as total body length of rats were analyzed by dual-energy X-ray absorptiometry using a DXA densitometer (UltraFocus^TM^ DXA System, Hologic, Tucson, AZ, USA). Total body length was measured from the nose to the end of the second caudal vertebra using the ruler provided by the software. All measurements were performed by the same technician. 

### 2.5. Micro-CT Analysis

Tibiae and vertebrae (LV4) were scanned ex vivo using a cabinet cone-beam Micro Compute Tomograph μCT 100 (SCANCO Medical AG, Brüttisellen, Switzerland). Tibia secondary spongiosa was scanned within the metaphysis below the growth plate and tibia cortical bone was scanned at midshaft. The LV4 spongiosa was scanned between the two growth plates. All scans were taken at 70 kVp, 114 μA using a tungsten target. Samples were scanned at 10 μm voxel-size with a 300 ms exposure time. Measured data were smoothed to partially suppress noise using a three-dimensional constrained Gaussian filter with finite filter support (1 voxel) and filter width (σ = 0.8). Images were later segmented to separate bone from the background using a global thresholding algorithm. A threshold of 430 mg HA/ccm was chosen for trabecular bone while a threshold of 520 mg HA/ccm was chosen for cortical bone. Regarding tibia growth plate, samples were segmented based on their grey-scale values in the CT slices and the region of interest was defined using the maximum fitted spheres method to compute growth plate thickness, surface, and volume [24] using the software provided by the equipment (IPL v6.0 (Image Processing Language, SCANCO Medical AG, Brüttisellen, Switzerland)).

Trabecular microarchitecture parameters including trabecular number (Tb.N, 1/mm), trabecular thickness (Tb.Th, mm), trabecular separation (Tb.Sp, mm), bone volume fraction (BV/TV, ratio), and connectivity density (Conn.D, 1/mm^3^) were determined. Cortical parameters including total cross-sectional area (Tt.Ar, mm^2^), cortical bone area (Ct.Ar, mm^2^), cortical area fraction (Ct.Ar/Tt.Ar, ratio), cortical thickness (Ct.Th, mm), cortical porosity (Ct.Sp), and the polar moment of inertia (pMOI, mm^4^), as a structural index of torsional resistance, were also determined.

### 2.6. Statistical Analysis

Normality of continuous variables was assessed using the Shapiro–Wilk test. Results were reported as mean ± SEM (standard error of mean). To evaluate differences between groups of study, a one-way ANOVA followed by post hoc analysis using Fisher’s protected least significant difference mean separation test was performed. When equal distribution of the variance was not achieved, a Brown–Forsythe and Welch ANOVA test was conducted instead. Whenever normal distribution or equal variance were not accomplished, Kruskal–Wallis tests were performed. To compared restricted and non-restricted groups during the restriction period a Student t-test or a Mann–Whitney U-test was performed. A *p*-value < 0.05 was considered statistically significant, while *p*-values between 0.05 and 0.10 were noted as a trend. Data treatment and analyses were performed using Graph Pad Prism 8 software (GraphPad Software Inc., San Diego, CA, USA). 

## 3. Results

### 3.1. Effect of Food Restriction on Body Weight, Length, Bone Densitometry, and Microstructure

The effect of the restriction period and the different experimental diets on body weight evolution was previously published in a recent publication of our group [20]. Body weight at the beginning of the experiment did not differ between experimental groups (NR: 89.4 ± 2.2 g; RDC: 88.3 ± 2.7 g; SDC: 91.6 ± 3.0 g). As expected, the restriction period exerted a significant reduction in the body weight gain of rats in the restricted group compared to NR group rats. Thus, the weight gain and, consequently, the growth of the restricted rats was slowed by the protein-calorie reduction of their intake. At the end of the restriction period, there was around a 56% of weight reduction in rats allocated to the restriction group (NR: 344.6 ± 5.6 g; RDC: 153.8 ± 4.4 g; SDC: 153.6 ± 5.3 g). 

The restriction period also affected the bone densitometry parameters producing a negative impact on BMD and BMC of axial and appendicular bones (vertebrae, tibia, and femur) (Appendix A). Similarly, most of the trabecular microarchitecture parameters evaluated were negatively influenced by the protein-calorie food restriction in both tibiae and vertebrae (Appendix A). In the same fashion, growth plate thickness, surface, and volume values of animals in the restricted group were notably lower compared to the reference values observed in the NR group after the nutritional restriction (Appendix A).

### 3.2. Effect of the Nutritional Intervention on Body Weight, Body Length, BMD, and BMC, during the Refeeding Period

Dietary intervention with the experimental diets induced a significant body weight increase in both RDC and SDC groups (around 80% increase compared to the body weight at the end of restriction), although they did not achieve to equal body weight of the NR group (NR: 408.6 ± 10.32 g; RDC: 312.0 ± 9.5 g; SDC: 303.4 ± 9.3 g). In this sense, both experimental groups exhibited similar food efficiency during the refeeding period (calculated as g gain (BW)/kcal consumed), as previously reported in Salto et al. [20].

Regarding total body length, none of the experimental groups completely recovered the values of total body length achieved by the NR group and no significant differences were found between experimental groups (NR: 227.4 ± 1.4 mm; RDC: 205.9 ± 2.0 mm *; SDC: 206.0 ± 2.8 mm *; * *p*-value < 0.05 versus NR group).

Bone densitometry parameters were differentially influenced by the nutritional interventions during the refeeding period. Both experimental groups improved BMD and BMC values of axial and appendicular bones, although they did not completely achieve the levels attained in the NR group at the same age. Between experimental groups, SDC group rats presented significantly higher BMD of the lumbar vertebrae 2–5 than the RDC group. Similarly, BMC of these bones was also positively impacted by the SDC diet, being BMC values 15.2% higher than those observed in the RDC group. Although no statistical significance was reached, BMD and BMC of appendicular bones (i.e., femurs and tibiae) of the SDC group rats were slightly higher than their respective in the RDC group (Figure 1).

### 3.3. Effects of the Nutritional Intervention on Trabecular and Cortical Microstructure 

Microstructure of isolated bones measured by micro-CT was in line with the pattern revealed by DXA analyses. Consequently, axial, and appendicular bones were positively modulated by the dietary intervention with SDC, exhibiting an improved trabecular microstructure compared to that of rats fed to RDC. In this sense, the tibia of SDC group rats presented significantly higher bone volume fraction (BV/TV), trabecular number (Tb.N), and connectivity density (Conn.D) along with a significantly lower trabecular separation (Tb.Sp) compared to NR and RDC groups (Figure 2) (Table 2). Notably, SDC diet induced a better recovery of the trabecular structure in the tibia, yielding more than 30% improved values in these parameters compared to the RDC diet. The analysis of the tibia midshaft cortical bone microstructure displayed a slight improvement in the SDC group compared to RDC, although none of the experimental groups attained the levels observed in the NR group, denoting than the restriction period produced an insult on cortical bone that was partially recovered by the nutritional intervention (Table 2).

In the same way, the SDC diet also exerted similar positive effects on the trabecular bone microstructure of the lumbar vertebra 4. Accordingly, the SDC group exhibited higher bone volume fraction (BV/TV), trabecular number (Tb.N), and connectivity density (Conn.D) along with lower trabecular separation (Tb.Sp) compared to NR and RDC groups. These parameters were at least +11% higher in the SDC group compared to RDC (Figure 3) (Table 2).

Not only was the trabecular bone of the vertebra positively impacted by the nutritional intervention but also the vertebra cortical bone displayed significant improvements. In this regard, the dietary intervention with the SDC diet produced an increase in vertebra cortical thickness (Ct.Th), cortical bone area (Ct. Ar), and cortical total cross-sectional area (Tt.Ar), which implies a significant improvement in comparison with the RDC group. Cortical porosity (Ct.Sp) was significantly lower in rats fed the SDC diet compared to RDC and NR (Table 2).

### 3.4. Effects of the Nutritional Intervention on Growth Plate

In line with the analysis of the trabecular and cortical microstructure, the analysis of the proximal tibia growth plate was also performed by micro-CT. Although SDC animals displayed slightly improved values compared to RDC, no statistical differences were found between them in terms of GP surface, thickness, or volume (Figure 4 and Figure 5). 

### 3.5. Effects of the Nutritional Intervention on Serum Metabolites

Postprandial levels of bone health-related hormones and metabolites at the time of sacrifice are presented in Table 3.

Although fasting glucose and one-hour postprandial insulin and glucose levels were not different between SDC and RDC, the insulin to glucose ratio was significantly higher in the RDC group compared to SDC and NR groups (as reported in Salto et al. [20]).

Regarding minerals, calcium, and magnesium levels did not differ between the groups of study. Phosphorus levels were significantly different in the ANOVA test (*p* < 0.001). However, no statistically significant differences in phosphorus levels were observed between RDC and SDC animals. 

Considering bone formation and resorption markers, refeeding led to significantly increased serum ALP levels, which is mainly accounted by the bone-specified ALP isoenzyme during growth [25,26], in animals subjected to a previous period of food restriction. Nonetheless, no differences were observed between SDC and RDC group animals after the refeeding period. In the same line, the levels of OPG after the refeeding period did not significantly differ between the groups of study. Likewise, no statistically significant differences in parathyroid hormone (PTH) levels were observed across the groups of study. Serum leptin levels measured one hour after oral gavage were significantly different across the groups of study (*p* = 0.025). In particular, higher leptin concentrations were observed in animals allocated to the RDC group compared to those allocated to the SDC group (*p* = 0.007). Furthermore, GH levels after the refeeding period were higher in the food-restricted animals compared to non-restricted rats. However, the type of experimental diet did not exert a significant effect on GH levels. On the other hand, LH levels did not significantly differ between any group of study. Finally, FSH levels were shown to vary significantly across groups of study with RDC group animals having higher FSH levels that those observed in NR and SDC groups (RDC vs. NR: *p* = 0.019; RDC vs. SDC: *p* = 0.020). 

## 4. Discussion

Undernutrition is a major cause of impaired lineal growth. When nutrition is re-established, a process of catch-up growth initiates with the purpose of restoring normal growth patterns. However, accelerated growth comes with impaired bone health, which has been associated with higher risk of bone fractures later in life [8,9]. In the present study, a diet containing slow digestible carbohydrates (SDC) was shown to improve parameters of bone mass and microarchitecture compared to rapid digestible carbohydrates (RDC) in a model of undernutrition-induced stunted catch-up growth rats.

We subjected weaning rats (approx. Day-21) to a four-week protein-calorie food restriction period (RR) receiving only 70% of the amount of food received by non-restricted rats (NR). At the end of this period, RR rats presented a lower growth rate, body length, and body weight compared to NR rats. Moreover, lower densitometry parameters (bone mineral density (BMD) and bone mineral content (BMC)) and impaired trabecular microarchitecture of axial and appendicular bones was observed in RR rats compared to NR animals. Food restriction during growth has been demonstrated to decrease BMD, BMC, and impair microarchitecture in different animal models [27]. Our results are in accordance with those obtained by Devlin et al., who fed 21-day mice 70% of the normal *ad libitum* consumption during six and twelve weeks observing a significantly lower whole-body BMD and BMC as well as impaired bone microstructure at both time points in restricted animals compared to mice fed *ad libitum* [28]. More prolonged restrictions (50% calorie intake for 70 days) have been associated with a significant reduction in BMD and weakened microarchitecture of tibiae also in rats [29]. On the other hand, different food restrictions have not achieved significant effects on bone parameters. For instance, Bryk et al., reported no differences in total skeleton BMD of rats after a four-week nutritional intervention with a low protein isocaloric diet (4% protein) since weaning [30]. Pando et al., reported that a 10 day-duration 40% calorie restriction diet from weaning did not significantly modify BMD of cortical bone [9]. Regarding humans, malnourished children exhibit not only growth retardation but also lower BMD and BMC due to a lack in bone mineralization during their development [31]. In the same way, adolescent patients suffering from anorexia nervosa, an eating disorder characterized by restricted food intakes, present decreased bone mass and quality denoted by a reduction in both BMD and BMC [32] along with poor cortical and trabecular microarchitecture, and increased risk of osteoporotic bone fractures [33,34]. 

Inadequate BMD and BMC during growth can lead to a suboptimal peak bone mass, which has been linked to an immediate increased risk of fractures during childhood and adolescence, as well as to long-term bone disorders such as osteoporosis [35,36,37]. In the present study, we observed that the refeeding period promoted growth and improved BMD and BMC values of food-restricted animals. In this sense, animals fed with the SDC diet during the refeeding period showed higher BMD and BMC values in appendicular and axial bones (i.e., femur, tibia, and vertebrae) compared to animals allocated to the diet containing the RDC blend. In our study, experimental animals aged ~79 days old average at the end of the refeeding period, which corresponds to the “young adulthood” stage of life and correlates to the age by which human epiphyseal closure is completed [38]. Hence, the increase in bone mineralization observed in the SDC group during the refeeding period may correlate to an increase in the peak bone mass at the end of the growth period in humans thus potentially decreasing the risk of osteoporosis later in life.

Moreover, the described SDC diet not only had a positive impact on bone macrostructure parameters but also improved the trabecular microstructure of vertebra and tibia compared to the RDC-containing diet. Specifically, analyzed tibiae and vertebrae of SDC group rats presented significantly higher bone volume fraction (BV/TV), trabecular number (Tb.N), and connectivity density (Conn.D) along with a significantly lower trabecular separation (Tb.Sp) compared to NR and RDC groups. Similarly, trabecular thickness (Tb.Th) of the tibia was significantly higher in SDC rats compared to RDC. Furthermore, the microCT analysis of the vertebra cortical microstructure revealed a similar pattern with SDC rats showing significant improvements in cortical thickness (Ct.Th), cortical bone area (Ct. Ar), and cortical total cross-sectional area (Tt.Ar) in comparison with NR and RDC groups. Nonetheless, tibia cortical microstructure was not differentially affected by the experimental diets. It is presumable that, due to the differential maturation of the cortical bone compared to trabecular bone, achieving significant improvements in cortical microstructure of longer bones might require a longer refeeding period. Notably, trabecular architecture is a determinant of bone strength and acts as a predictor of fractures related to bone fragility in sites such as vertebrae and femoral head [39]. Given that humans vertebrae are more likely to suffer osteoporosis-related fractures, a greater cortical and trabecular microstructure of the spine could be helpful for the prevention of these fractures in elder individuals [40].

Two potential mechanisms could be responsible for these positive results observed in bone features of the SDC group. One mechanism would be represented by an improved control of the insulin-glycemic response. In animals, catch-up growth was demonstrated to promote insulin resistance through dysfunction of the adipose tissue [41]. In addition, it has also been shown that hyperinsulinemia, together with an impaired control of the blood glucose levels, is associated with decreased bone mass and defective bone microarchitecture [14,42]. In this sense, studies conducted in streptozotocin-induced diabetic adult rats, which are characterized by induced hyperglycemia, have observed that diabetic animals display lower BMD and connection density along with compromised bone microstructure compared to control healthy rats [14]. Similarly, another study evaluating the effects of prenatal and postnatal dietary manipulations on bone parameters of rats concluded that a perinatal high fat diet was associated with acquisition of suboptimal skeletal parameters which was mediated by elevated insulin levels [43]. In humans, nutritional restriction during the early stages of life followed by a period of catch-up growth during infancy promotes a higher insulin response relative to that of children with a normal growth pattern [44]. Remarkably, diabetic children tend to present low BMD and decreased circulating IGF-1 levels, although associations between diabetes and bone health are typically heterogeneous and rely on diverse bone turnover biomarkers [45]. We previously reported that diets containing the described SDC blend promote a healthier catch-up growth characterized by decreased insulin resistance in terms of blood insulin/glucose ratio, as well as improved muscle function and fuel utilization [20], thus counteracting the negative consequences of catch-up growth. Hence, the improved control of postprandial glucose peaks elicited by the SDC diet in contraposition to the RDC diet during catch-up growth supports a healthier bone growth and bone development mediated by the prevention of the negative effect that impaired glucose and insulin control has on BMD, BMC, and microstructure. As previously commented, unhealthy catch-up growth correlates to higher risk of metabolic diseases, such as diabetes and obesity, later in life [7]. Moreover, diabetic, and obese adult patients are at higher risk of bone fractures due to impaired bone microstructure [42,46]. Overall, preventing the development of a thrifty phenotype characterized by impaired insulin sensitivity might exert short- and long-term beneficial effects on bone health parameters (i.e., BMD and bone trabecular and cortical microstructure) of children undergoing catch-up growth, leading to a higher peak bone mass at the end of their adolescence. 

A second mechanism would be mediated by the prebiotic effect exerted by the SDC diet. Fructo-oligosaccharides (FOS), inulin enriched FOS, and IMOs are considered to exert prebiotic effects modulating gut microbiota which leads to the production of short chain fatty acids (SCFA), such as butyric, propionic, and acetic acids [47,48,49]. SCFA are associated with enhanced intestinal mineral absorption in vivo through a decrease in the pH of the cecum [50,51]. In this regard, consumption of foods rich in calcium and calcium supplementation in children increase bone mineral acquisition [52] while providing a combination of calcium with other micronutrients, such as magnesium, iron, or zinc, might potentially enhance growth in stunted children [53]. Enhanced mineralization may support the improvements on BMD, BMC, and trabecular microstructure observed in SDC compared to the RDC group. In line with our results, it has been reported that the administration of FOS and FOS-enriched inulin to rats has a direct effect on bone, increasing both micro (cortical and trabecular parameters) and macrostructure (BMD and BMC) in processes of growth, development, and osteoporosis [54]. Similarly, a recent study reported that the administration of isomaltulose elevates the relative abundance of *Actinobacteria,* and notably increases the production of SCFA, particularly propionic acid and, to a lesser extent, butyric acid [55]. Propionate and butyrate have been shown to modulate bone homeostasis by increasing bone mass and decreasing bone resorption [56,57,58]. Curiously, a recent report suggested that SCFAs mediate the anabolic effects of the intermittent administration of PTH on bone turnover [59]. 

In our study, PTH levels were around 29% lower in the SDC group compared to RDC which is indicative of lower bone mineral resorption [60]. On the other hand, serum alkaline phosphatase (ALP), an enzyme that increases local phosphorus concentrations and promotes hydroxyapatite deposition in the skeletal matrix [61], was shown to be significantly increased in restricted animals after the refeeding period. During bone growth, serum ALP activity is elevated, mainly due to the bone-specific ALP isoenzyme [26], which is related to bone formation. Among causes of decreased ALP activity are the cessation of bone growth and the multi-nutritional deficiency of minerals such as magnesium and zinc [25]. Thus, food restriction leads to decreased serum ALP levels, which in turn increase during catch-up growth as a sign of an enhanced bone mineralization process [9]. Overall, the SDC group presented around 10% higher serum ALP and 14% higher OPG levels (both markers of bone formation) than those observed in the RDC group, which, in combination with the lower PTH levels, denotes improved bone mineralization, as displayed by the increase in BMD and BMC, as well as the higher trabecular parameters in axial and appendicular bones of the SDC group. 

In humans, stature is a consequence of the axial longitudinal growth of long bones, which takes place during infancy, and progressively decelerates until eventually ceasing during young adulthood [62]. Longitudinal bone growth ceases due to epiphyseal fusion which is the result of a progressive process of senescence that leads to the replacement of the growth plate (GP) cartilage by bone tissue [63,64]. Although rats maintain the epiphyseal plate until old age [65], higher GP surface, thickness, and volume imply a more active GP, which leads to a higher margin for longitudinal growth. Food restriction attenuates longitudinal growth, and catch-up growth gradually reverses these effects [66]. In the present study, we observed that the consumption of the SDC diet was linked to a modest improvement in growth plate parameters. Specifically, all three growth plate parameters displayed higher values in the SDC compared to the RDC group, from 4% in GP surface to 8% in GP volume. The lack of statistical significance in these results might be consequence of the period of study evaluated, as rats received the nutritional intervention from childhood to adolescence, a period characterized by lower GP activity due to GP maturation [67]. As depicted in Appendix A, GP surface and thickness undergo profound changes from weaning to childhood in non-restricted rats, while these differences are far less pronounced from childhood to later stages of life. Therefore, a narrower margin for GP development might explain the modest improvements observed in GP parameters of SDC rats assessed during young adulthood. 

Furthermore, we observed that growth hormone (GH) levels were higher in both SDC and RDC animals compared to the NR group at the end of the refeeding period, which implies an active process of growth in animals previously subjected to food restriction. Similarly, leptin is a bone-related systemic factor strongly influenced by the nutritional status that might mediate this association [9]. Sex hormones are the main responsible for GP fusion, and leptin has been observed to stimulate aromatase expression and synthesis, as well as estrogen receptor expression in vitro [68]. Interestingly, high leptin levels have been associated with a loss of pubertal growth spurt in obese compared to non-obese children, leading to a lower final stature [68]. SDC diet showed markedly lower postprandial leptin levels compared to NR and RDC groups. These results might support a positive effect of a nutritional intervention with the described SDC blend on the growth plate of stunted children, which would lead to an improved recovery of linear growth. 

We also analyzed other potential biomarkers of bone health. Follicle-stimulating hormone (FSH) has been demonstrated to exert a direct effect on bone, increasing the activity of osteoclast stimulating bone resorption in an estrogen-independent manner [69], and high levels are linked to visceral obesity [70]. Accordingly, blocking the action of FSH has been proposed to treat obesity and osteoporosis concurrently [70]. Rats allocated to the SDC diet displayed FSH levels comparable to those of the NR group, and significantly lower levels than those observed in the RDC diet group. Overall, biochemical analyses showed that SDC diet exerted a more pronounced positive effect on bone biomarkers profile of SDC rats compared to the RDC diet, which translated into improved bone parameters in the SDC group. Finally, further studies should evaluate the influence of SDC on gut microbiota and SCFAs production in order to provide insights into underlying mechanisms of action. 

Based on the results obtained in the present study and our previous research [20], SDC consumption was proved to improve wide aspects of musculoskeletal health by promoting an adequate bone formation characterized by increased bone mass and quality, in terms of BMD, BMC, and microarchitecture, while also improving features of muscle function and muscle fuel utilization during catch-up growth. 

## 5. Conclusions

A diet containing slow digestible carbohydrates elicited significant improvements in bone densitometry parameters and the bone microstructure of rats undergoing catch-up growth after a transient period of food restriction compared to a diet containing rapid digestible carbohydrates. Slow digestible carbohydrates might be useful to promote a healthier catch-up growth musculoskeletal status in children, promoting a higher peak bone mass that could exert a greater protection against the bone loss that occurs later in life. 

## Figures and Tables

**Figure 1 nutrients-14-01303-f001:**
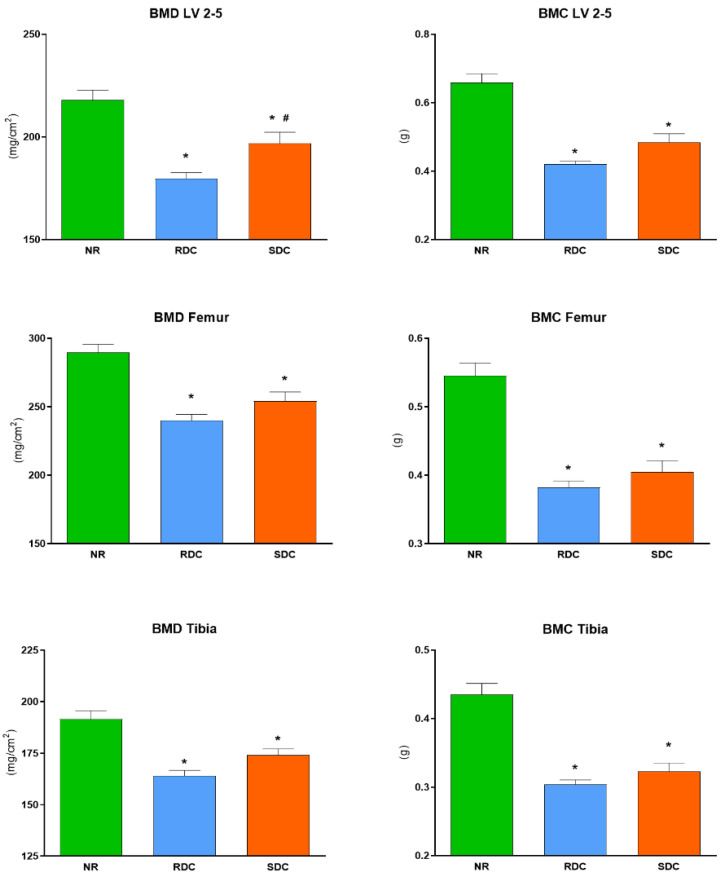
DXA analysis of bone mineral density (BMD) and content (BMC) at the end of the refeeding period. Values are expressed as mean ± S.E.M. NR group, not restricted group; RDC group, refeeding with rapid digestible carbohydrates diet; SDC group, refeeding with slow digestible carbohydrates diet; LV, lumbar vertebrae. * Significant difference compared to NR group (*p*-value < 0.05); # significant difference compared to RDC group (*p*-value < 0.05).

**Figure 2 nutrients-14-01303-f002:**
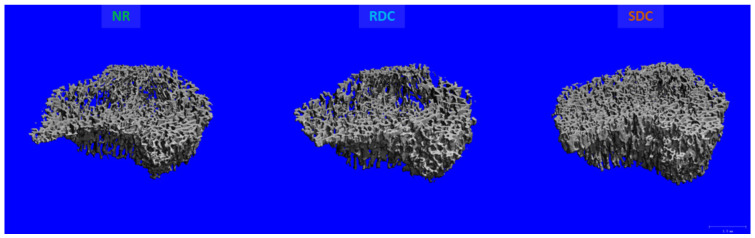
Representative MicroCT images of the tibia trabecular bone after the refeeding period. NR group, not restricted group; RDC group, refeeding with rapid digestible carbohydrates diet; SDC group, refeeding with slow digestible carbohydrates diet.

**Figure 3 nutrients-14-01303-f003:**
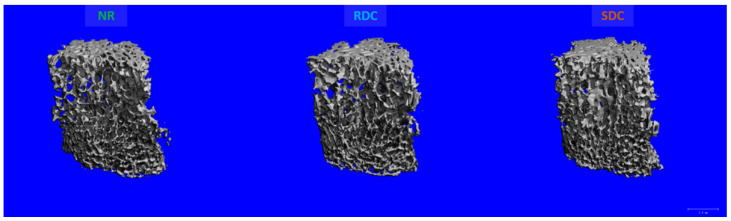
Representative MicroCT images of the vertebra 4 trabecular bone after the refeeding period. NR group, not restricted group; RDC group, refeeding with rapid digestible carbohydrates diet; SDC group, refeeding with slow digestible carbohydrates diet.

**Figure 4 nutrients-14-01303-f004:**
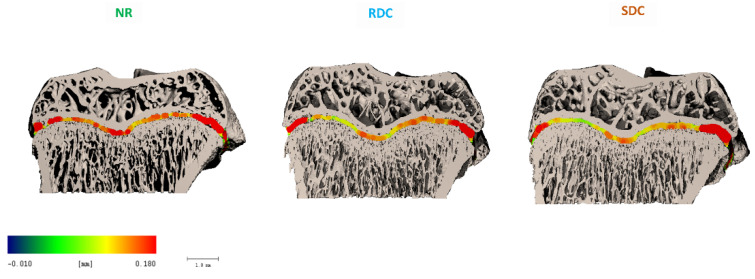
Representative micro-CT images of the proximal tibia GP after the refeeding period. Colors represent epiphyseal GP height in mm. NR group, not restricted group; RDC group, refeeding with rapid digestible carbohydrates diet; SDC group, refeeding with slow digestible carbohydrates diet.

**Figure 5 nutrients-14-01303-f005:**
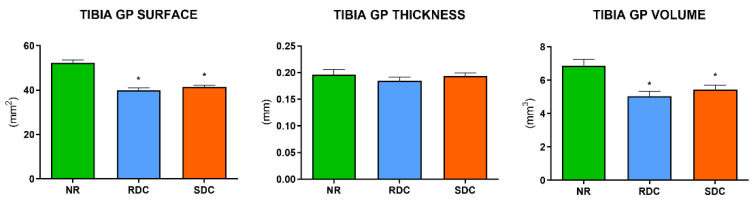
Micro-CT analysis of tibia growth plates at the end of the refeeding period. Values are expressed as mean ± S.E.M. NR group, not restricted group; RDC group, refeeding with rapid digestible carbohydrates diet; SDC group, refeeding with slow digestible carbohydrates diet; GP, growth plate. * Significant difference compared to NR group (*p*-value < 0.05).

**Table 1 nutrients-14-01303-t001:** Composition of control (NR) and experimental diets (RDC and SDC).

Ingredients	AIN93M	RDC	SDC
	NR	RDC	SDC
CHO (g/100 g diet)	78.5	67.0	67.0
Sucrose (g/100 g CHO)	13.0	32.0	
Isomaltulose (g/100 g CHO)			26.4
Sucromalt^®^ (g/100 g CHO)			22.1
Cornstarch (g/100 g CHO)	61.0		
MDs (g/100 g CHO)	20.0	64.0	23.0
IMOs (g/100 g CHO)			11.5
Resistant MD (g/100 g CHO)			10.0
FOS or Inulin enriched FOS (g/100 g CHO)		4.0	7.0
Cellulose (g/100 g CHO)	5.0		
Protein (g/100 g diet)	14.0	15.0	15.0
Fat (g/100 g diet)	4.0	9.8	9.8

Sucromalt^®^ consists of a slow digestive combination of sucrose and maltose. NR group, not restricted group; RDC group, refeeding with rapid digestible carbohydrates diet; SDC group, refeeding with slow digestible carbohydrates diet; CHO, carbohydrates; FOS, fructooligosaccharides; IMOs, isomalto-oligosaccharides; MD, maltodextrins.

**Table 2 nutrients-14-01303-t002:** Micro-CT trabecular and cortical parameters of tibia and vertebra after refeeding.

Micro-CT Parameter	Tibia			Vertebra		
	NR	RDC	SDC	NR	RDC	SDC
BV/TV (ratio)	0.124 ± 0.010	0.118 ± 0.005 *	0.164 ± 0.026 *^#^	0.261 ± 0.007	0.265 ± 0.003	0.307 ± 0.014 *^#^
Tb.Th (mm)	0.064 ± 0.001	0.054 ± 0.001 *	0.051 ± 0.001 *^#^	0.076 ± 0.001	0.073 ± 0.001 *	0.072 ± 0.001 *
Tb.N (1/mm)	2.484 ± 0.215	2.765 ± 0.179 *	4.425 ± 0.305 *^#^	3.448 ± 0.072	3.667 ± 0.168	4.064 ± 0.116 *^#^
Tb.Sp (mm)	0.417 ± 0.039	0.374 ± 0.028 *	0.208 ± 0.013 *^#^	0.269 ± 0.006	0.251 ± 0.004 *	0.226 ± 0.008 *^#^
Conn.D (1/mm^3^)	59.52 ± 6.703	71.31 ± 4.068 *	141.9 ± 14.10 *^#^	69.62 ± 2.901	89.62 ± 3.359 *	111.3 ± 6.682 *^#^
Ct.Th (mm)	0.576 ± 0.007	0.508 ± 0.008	0.515 ± 0.010	0.259 ± 0.006	0.221 ± 0.005 *	0.242 ± 0.006 *^#^
Tt.Ar (mm^2^)	4.749 ± 0.115	3.641 ± 0.074 *	3.686 ± 0.102 *	3.103 ± 0.08	2.400 ± 0.054 *	2.661 ± 0.104 *^#^
Ct.Ar (mm^2^)	4.687 ± 0.103	3.609 ± 0.073 *	3.660 ± 0.102 *	3.007 ± 0.078	2.292 ± 0.058 *	2.555 ± 0.105 *^#^
Ct.Ar/Tt.Ar (ratio)	0.993 ± 0.001	0.991 ± 0.001	0.993 ± 0.001	0.970 ± 0.001	0.9577 ± 0.003 *	0.963 ± 0.001 *
pMOI (mm^4^)	7.216 ± 0.352	4.344 ± 0.170 *	4.423 ± 0.264 *	-	-	-

Data presented as mean ± S.E.M. NR group, not restricted group; RDC group, refeeding with rapid digestible carbohydrates diet; SDC group, refeeding with slow digestible carbohydrates diet; BV/TV, bone volume/total volume; Tb.Th, trabecular thickness; Tb.N, trabecular number; Tb.Sp, trabecular separation; Conn.D, connectivity density; Ct.Th, cortical thickness; Tt.Ar, cortical total cross-sectional area; Ct.Ar, cortical bone area; pMOI, polar moment of inertia. * Significant difference compared to NR group (*p*-value < 0.05); # significant difference compared to RDC group (*p*-value < 0.05).

**Table 3 nutrients-14-01303-t003:** Bone health-related hormones and serum metabolites.

Metabolite	Not Restricted Group	RDC Group	SDC Group
Calcium (mmol/L)	2.443 ± 0.020	2.461 ± 0.018	2.467 ± 0.111
Magnesium (mg/dL)	2.181 ± 0.081	2.260 ± 0.090	2.327 ± 0.065
Phosphorus (mg/dL)	5.388 ± 0.111	7.122 ± 0.418 *	6.525 ± 0.211 *
ALP (UI/L)	110.4 ± 6.696	160.5 ± 16.140 *	175.6 ± 6.793 *
PTH (pg/mL)	28.81 ± 4.524	45.85 ± 8.638	34.27 ±7.557
OPG (pg/mL)	716.7 ± 67.14	618.8 ± 39.04	705.5 ± 30.52
GH (pg/mL)	2957 ± 957.8	9327 ± 1840.0 *	8162 ± 2.152 *
Insulin (pg/mL)	662.1 ± 138.9	787.1 ± 159.9	589.7 ± 148.1
Leptin (pg/mL)	5413 ± 618.6	6738 ± 790.7	3855 ± 620.0 ^#^
LH (pg/mL)	505.5 ± 88.41	730.0 ± 127.7	644.0 ± 111.1
FSH (pg/mL)	6635 ± 426.4	8077 ± 481.7 *	6607 ± 307.3 ^#^

Data presented as mean ± S.E.M. RDC, rapid digestibility carbohydrates; SDC, slow digestibility carbohydrates; ALP, alkaline phosphatase; PTH, parathyroid hormone; OPG, osteoprotegerin; GH, growth hormone; LH, luteinizing hormone; FSH, follicle-stimulating hormone. * Significant difference compared to NR group (*p*-value < 0.05); # significant difference compared to RDC group (*p*-value < 0.05).

## Data Availability

The data presented in this study are available on request from the corresponding author. The data are not publicly available due to privacy reasons.

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
