# Peer review of "Dietary Complex and Slow Digestive Carbohydrates Promote Bone Mass and Improve Bone Microarchitecture during Catch-Up Growth in Rats"

_nutrients, 2022, doi:10.3390/nu14061303_

Round 1
Reviewer 1 Report
I have read carefully the manuscript entitled “Dietary Complex and Slow Digestive Carbohydrates Promote Bone Mass and Improve Bone Microarchitecture During Catch-Up Growth in Rats” and although some interesting results can be observed, Authors didn't draft the paper well. The Authors should clarify some issues regarding the manuscript before the re-submission of the work to Nutrients.
First at all, I believe that the experimental diet should be placed int the manuscript. This is a crucial information for the whole experiment and must be directly given to the readers.
Figures 3, 5 and 6 repeat the data from table 1 and therefore should be removed.
Verify the results for BV/TV and Tb.N - Just about 0.1 % of bone fraction in bone tissue? Shouldn’t it be about 10%? Similarly, only 2-3 trabeculae per mm with connectivity density of the range of 100/mm3 ?
The results in 3.5 section are presented in a different manner as the previous ones - more detailed statistical data are induced (CI, F-statistic, exact p-values). However, to maintain the uniformity throughout the manuscript, please remove these data or supplement the previous results with similar ones.
What is the source of data presented in Fig S1 ?
L81-82 Results of grip strength measurements are indeed presented in [2] but the methods of measurement procedures are missing. Therefore, I suggest removing this part as these results cannot be verified/rerepeated by other researchers.
L98 Since 2011, all experiments involving animas should be conducted in accordance with Directive 2010/63/EU Of The European Parliament And Of The Council.
L119 (femora, tibiae, lumbar vertebrae)
L129 As in this part of the experiment only selected bones and blood serum were analyzed, the information that the other tissues were also collected is unnecessary.
L164-174 What software was used for all these measurements ?
L168-173 superscripts.
L175 I must admit that I don’t fully understand performed statistical analysis. As ANOVA with Fisher's LSD test was performed between 3 groups, it is not clear why also t-test was performed. I assume that it was used to compare two restricted groups. Why Authors did not perform contrast analysis between these groups ?
L188-200 this is the summary of methods section, it does not show any results. Please consider removing the whole paragraph.
L215-216 “NR groups animals displayed normal 215 growth plate parameters for their age” - reference needed.
L226 Why feed intake data is not presented?
L237-240 Rephase this sentence as no statistical significance was reached.
Supplementary table S1 doesn’t show parameters of cortical bone. In Tables S1 and S3 the units of presented parameters are missing.
Figure 7 - What do these colors represent? The legend is missing.
L244-345 and L481-486 In this study, a general ALP was measured, not bone-specific isoform of alkaline phosphatase (BALP).It cannot therefore concluded that observed results are mainly due to increase of BALP. Please rephase.
L417-420 not clear enough – Did Pando et al also use SDC and RDC diets in their work ?
L422-424 no such statement can be found in [35]. Bone microarchitecture can be used as an additional determinant of fracture risk, especially in vertebra or femoral head. However, in long bones (tibia) bone fragility also appears in bone midshaft which lacks of trabecular bone.
Full bibliographic data for [58] is missing.
Reviewer 2 Report
In this manuscript, the authors evaluated the effects of a diet containing slow (SDC) and rapid (RDC) digestible carbohydrates on bone quality parameters during the catch-up growth period in a model of diet-induced stunted rats. They found that the SDC diet was shown to improve BMD and BMC of appendicular and axial bone, and have positively affected the trabecular microarchitecture of tibiae and vertebrae after a four-week refeeding period in comparison with the RDC diet. The authors suggested that two potential mechanisms could be responsible for SDC diet supports healthier bone growth and bone development during catch-up growth: SDC diet prevents the development of a thrifty phenotype characterized by impaired insulin sensitivity; SDC diet mediates the prebiotic effects by modulating gut microbiota to enhance intestinal mineral absorption. This study and the authors' previous study suggest that slowly digestible carbohydrates are better than rapidly digestible carbohydrates might be useful to promote a healthier catch-up growth musculoskeletal status in children. The findings are significant, the data convincing and this manuscript is easy to follow.
no comment.
Round 2
Reviewer 1 Report
Thanks to the authors for their diligent attention to the comments made. For my part, I believe that the article has been substantially improved, so that it can be published in its present form.